# Real-World Presentation and Prognostic Effect of Allogeneic Blood Transfusion during the Intensive Induction Phase in Pediatric Acute Lymphoblastic Leukemia

**DOI:** 10.3390/cancers15184462

**Published:** 2023-09-07

**Authors:** Kunyin Qiu, Xiongyu Liao, Yang Li, Ke Huang, Honggui Xu, Jianpei Fang, Dunhua Zhou

**Affiliations:** 1Department of Hematology/Oncology, Children’s Medical Center, Sun Yat-sen Memorial Hospital, Sun Yat-sen University, Guangzhou 510120, China; qiuky@mail2.sysu.edu.cn (K.Q.); liaoxy7@mail.sysu.edu.cn (X.L.); liyang5@mail.sysu.edu.cn (Y.L.); hke@mail.sysu.edu.cn (K.H.); xuhgui@mail.sysu.edu.cn (H.X.); 2Guangdong Provincial Key Laboratory of Malignant Tumor Epigenetics and Gene Regulation, Sun Yat-sen Memorial Hospital, Sun Yat-sen University, Guangzhou 510120, China

**Keywords:** ABT, ALL, FFP, children, prognostic

## Abstract

**Simple Summary:**

In China, approximately 15,000 cases of childhood acute lymphoblastic leukemia (ALL) are diagnosed each year, and it is also the most common hematological cancer and the leading cause of tumor death under the age of 18. At present, ALL remains one of the major indications for allogeneic blood (ABT). This study aims to determine associations between ABT during the intensive induction phase of therapy and prognostic effect in a real-world cohort of pediatric patients with ALL. We found that among the blood products, only fresh frozen plasma (FFP) infusion is closely related to the prognosis of childhood ALL. During the intensive induction phase, the indications of FFP transfusion should be strictly grasped, and the total amount of FFP should be controlled and kept below 25 mL/kg.

**Abstract:**

**Purpose**: To determine associations between allogeneic blood transfusion (ABT) during the intensive induction phase of therapy and prognostic effect in a real-world cohort of pediatric patients with acute lymphoblastic leukemia (ALL). **Methods**: A total of 749 pediatric patients who were diagnosed with ALL were enrolled in this study by using a single-center retrospective cohort study method from February 2008 to May 2022. **Results**: Among the ABT patients, 711 (94.9%) children were transfused with packed red blood cells (PRBCs), 434 (57.9%) with single-donor platelets (SDPs), and 196 (26.2%) with fresh frozen plasma (FFP). Our multivariate analysis demonstrated that FFP transfusion was the unique independent factor that affected both relapse-free survival (RFS) and overall survival (OS). The transfusion of FFP was significantly associated with higher age (*p* < 0.001), being more likely to receive SCCLG-ALL-2016 protocol (*p* < 0.001), higher proportion of more than 25 blood product transfusions, more PRBC transfusion (*p* < 0.001), and higher D33-MRD-positive rates (*p* = 0.013). Generalized additive models and threshold effect analysis using piece-wise linear regression were applied to identify the cut-off value of 25 mL/kg for average FFP transfusion. K-M survival analysis further confirmed that average FFP transfusion > 25 mL/kg was an independent adverse indicator of inferior outcome in terms of RFS (*p* = 0.027) and OS (*p* = 0.033). **Conclusions**: In blood products, only FFP supplement is closely related to the prognosis of childhood ALL. During the intensive induction phase, the indications of FFP transfusion should be strictly grasped, and the total amount of FFP should be controlled and kept below 25 mL/kg.

## 1. Introduction

Allogeneic blood transfusion (ABT) exposes the recipient to a large number of soluble and cell-related forms of alloantigens and can create conditions for a variety of possible immunological responses, including alloimmunization and down-regulation of the immune response. In the literature of transfusion medicine, the complex of immune changes related to transfusion is known as transfusion-related immunomodulation (TRIM) effects [1,2]. TRIM was initially discovered to have a beneficial effect by increasing the survival of patients with renal allografts [3]. However, the effect of TRIM was later associated with tumor growth, bacterial degradation, postoperative mortality, and organ dysfunction [4,5,6,7].

In China, approximately 15,000 cases of childhood acute lymphoblastic leukemia (ALL) are diagnosed each year, and it is also the most common hematological cancer and the leading cause of tumor death under the age of 18 [8]. At present, ALL remains one of the major indications for ABT [9]. Previous studies have shown that most children with ALL require ABT during treatment, mainly during the induction phase [10,11]. It has been proposed that TRIM effects during this critical induction period might adversely contribute to ALL patient outcomes through immune suppression [11,12,13]. However, some scholars also suggest that TRIM is unlikely in childhood ALL and that the poor outcome associated with transfusions is likely a secondary effect due to disease severity and reduction in chemotherapy doses [10,14].

In general, studies of TRIM and the prognosis of pediatric ALL are still few and controversial [10,11,12,13,14]. The purpose of the present study is to determine associations between ABT during the intensive induction phase of therapy and prognostic effect in a real-world cohort of pediatric patients with ALL.

## 2. Patients and Methods

### 2.1. Study Design

This is a single-center, real-world, retrospective cohort study. Patients were regarded as eligible when meeting the following criteria: (1) age ≤ 18 years; (2) clinical presentation consistent with ALL and diagnosis of ALL based on morphological review of bone marrow smears, immunophenotyping, cytogenetics, and molecular genetics according to the WHO 2008 criteria; (3) first-episode children. Patients were excluded for the following reasons: (1) mature B; (2) secondary to immunodeficiency disease; (3) as a second malignancy; (4) Down’s syndrome; (5) glucocorticoid use for more than 1 week in the month before enrollment; (6) patients missing ABT data.

### 2.2. Study Participants

Finally, a total of 749 child patients who were diagnosed with ALL were recruited for this study from 28 February 2008 to 31 May 2022. Patients were then classified into two groups according to whether fresh frozen plasma was transfused or not. The study was conducted in accordance with the principles set down in the Declaration of Helsinki and was approved by the Ethics Committee of Sun Yat-sen Memorial Hospital. All patients, or the patients’ guardians, provided written informed consent.

### 2.3. Chemotherapy Protocol and Treatment Response

Children diagnosed between February 2008 and September 2016 were treated according to the Guangdong Children’s Leukemia Group-ALL-2008 (GD-ALL-2008) protocol [15], and children diagnosed between October 2016 and May 2022 were treated according to the South China Children’s Leukemia Group-ALL-2016 (SCCLG-ALL-2016) protocol [16]. The chemotherapeutic drug classes and the composition of the chemotherapy protocol in the intensive induction phase were essentially the same for both regimens, i.e., diagnosis and assessment of sensitivity after 7 days of pretreatment with prednisone upon enrollment, continued initiation of VDLD (vincristine + dexamethasone + L-asparaginase (L-ASP) + daunorubicin) according to the risk stratification criteria, and three triple intrathecal injections to prevent central nervous system (CNS) leukemia. During this period, a bone marrow morphology examination was performed on Day 15 and Day 33 adjust the risk level dynamically, and dexamethasone was decreased to a halt within 29 to 38 days. The risk stratification includes low, intermediate, and high risks according to our previous chemotherapy protocol [15,16].

Early response to treatment was measured as the absolute number of peripheral lymphoblasts on Day 8 of the induction phase. Patients were classified as prednisone good responders (PGRs) when the absolute peripheral lymphoblast count by induction Day 8 was less than 1000/μL and as prednisone poor responders (PPRs) when the count was 1000/μL or higher. Minimal residual disease (MRD) and bone marrow (BM) morphology were performed on Day 15 and Day 33, and patients were classified according to their blast cell amount as M1 (blast cells < 5%), M2 (5% to <25%), or M3 (>25%). MRD-positive on Day 15 was defined as MRD ≥ 0.1%, while MRD-positive on Day 33 was defined as MRD ≥ 0.01%.

### 2.4. Transfusion Events

The blood products were provided by Guangzhou Blood Center. Various blood components were obtained through routine procedures from donors and were treated using leukapheresis without γ-laser radiation. The number, frequency, time, and type of any blood product transfusion and its preparation method were recorded for all patients in an electronic blood transfusion database. During the intensive induction phase, the transfusion of packed red blood cells (PRBCs), single-donor platelets (SDPs), or fresh frozen plasma (FFP) was considered as a transfusion event. In detail, for patients with hemoglobin <70 g/L, platelet counts <20 × 10^9^/L, or active hemorrhage, PRBCs and SDPs were transfused, respectively, while for those with fibrinogen <1.5 g/L or abnormal coagulation function, FFP was selectively transfused. Other transfusion events such as cryoprecipitate or blood-derived platelet concentrate were negligible and not considered for analysis. The decision making of ABT was based on the pediatric physicians’ clinical judgment and was not completely limited by the above test thresholds.

### 2.5. Follow-Up

All cases were followed up by outpatient review or telephone, with follow-up dates up to 30 June 2016 for children receiving the GD-ALL-2008 protocol and up to 30 June 2022 for children receiving the SCCLG-ALL-2016 protocol, with study endpoints set as death, lost to follow-up, or follow-up cutoff, and up to follow-up time for those lost to follow-up. The relapse-free survival (RFS) and overall survival (OS) of the study cohort were analyzed. RFS was defined as the duration from diagnosis until relapse, and the patients without any events at the time of the final follow-up were censored. OS was defined as the duration from diagnosis to death, and the patients who remained alive at the final follow-up were censored.

### 2.6. Statistical Analyses

Descriptive statistics were used to summarize variables related to the demographics and clinical characteristics of the patients. Groups were compared using Fisher’s exact test as appropriate for categorical variables and the Kruskal–Wallis test for continuous variables. The probabilities of RFS and OS were estimated according to the Kaplan–Meier method, and univariable comparisons among the groups were performed using the log-rank test. The Cox proportional hazard model was used for multivariate analysis of RFS and OS. The results are expressed as hazard ratios and their 95% confidence intervals (CIs). In order to explore whether there was a non-linear relationship between FFP transfusion and risk of relapse, after adjusting for all covariates, the correlation fitting curve between FFP transfusion and risk of relapse was drawn using the restrictive cubic spline function following the Cox proportional hazards models. Threshold levels (i.e., turning points) were determined by trial and error, which involves selecting turning points along a predetermined interval and then choosing turning points. All tests were two-sided, and a *p* value of <0.05 was considered to indicate statistical significance. All statistical analyses were performed using SPSS statistical software version 22.0 and EmpowerStats 2.0 (http://www.empowerstats.cn/ (accessed on 7 March 2023)).

## 3. Results

### 3.1. Baseline Characteristics of the Study Population

In total, 749 patients were enrolled in the present study, including 462 males and 287 females, with a median age of 4.4 years (range: 0.4 to 14.9). Around 54.6% (*n* = 409) of patients were treated with the GD-ALL-2008 protocol and 45.4% (*n* = 340) of patients with the SCCLG-ALL-2016 protocol. The immunophenotype in most of the patients was B-cell type (90.9%), followed by T-cell (7.9%) and biphenotypic leukemia (1.2%) in 59 patients and 9 patients, respectively. Baseline characteristics of the study population are summarized in Table 1. During the critical induction period of chemotherapy, 724 (96.6%) pediatric ALL patients received ABT. The median number of transfusions was 6.0 (mean, 6.7; range: 0–40). Among the ABT patients, 711 (94.9%) children were transfused with PRBCs, 434 (57.9%) with SDPs, and 196 (26.2%) with FFP. The median number of PRBC and SDP transfusions for each was 4 (mean, 2; range, 0.0–25.5) and 1 (mean, 3.8; range, 0.0–20.0), respectively.

The median age of patients with FFP transfusion was higher than those without FFP transfusion (5.7 years vs. 4.1 years, *p* < 0.001), and FFP transfusion was far more common in older patients (≥10 years) (23.5% vs. 12.5%, *p* < 0.001). Within the cohort receiving the SCCLG-ALL-2016 protocol, patients with FFP transfusion had a significantly lower prevalence than patients without FFP transfusion (17.3% vs. 55.3%; *p* < 0.001). More than 25 blood product transfusions was mostly observed in the cohort with FFP transfusion than those without FFP (3.1% vs. 0.5%, *p* < 0.001). Patients with FFP transfusion were accompanied by more PRBC transfusions than those without FFP transfusion (4 U vs. 3 U; *p* < 0.001). On Day 33 of induction therapy, participants with FFP transfusion in the M1 bone marrow category seemed to have a lower proportion when compared with those participants without FFP transfusion (94.6% vs. 97.3%, *p* = 0.085). Importantly, the prevalence of MRD-positivity among children with FFP transfusion was higher than that among those without FFP transfusion (27.5% vs. 16.5%, *p* = 0.013) on Day 33.

### 3.2. Univariate Analysis for RFS and OS among Pediatric ALL Patients

In univariate analysis, the known prognostic factors among pediatric ALL patients were examined, as summarized in Table 2. The SCCLG-ALL-2016 protocol and PLT ≥ 20 × 10^9^/L were significantly predictive of better RFS, while WBC ≥ 50 × 10^9^/L and D15-BM M2/M3 revealed a significant decrease in RFS. In addition, age ≥ 10 years, WBC ≥ 50 × 10^9^/L, high-risk group, T cell phenotype, D15-BM M2/M3, and D33-BM M2/M3 had a significant association with poor OS. In terms of ABT, PRBC, SDP, and FFP transfusions had a significant association with adverse outcome. For PRBCs, the HRs for RFS and OS were 1.1 (95%*CI*, 1.0–1.1, *p* = 0.004) and 1.1 (95%*CI*, 1.0–1.2, *p* < 0.001), respectively. For SDPs, the HRs were 1.1 (95%*CI*, 1.0–1.1, *p* = 0.087) and 1.1 (95%*CI*, 1.0–1.2, *p* = 0.002) for RFS and OS, respectively. Also, FFP transfusion was significantly associated with poor RFS (*HR* = 1.5, 95%*CI*:1.1–2.4, *p* = 0.047) and OS (*HR* = 2.4, 95%*CI*:1.4–3.9, *p* < 0.001). No differences were found between transfusion times and RFS or OS.

### 3.3. Multivariate Analysis for RFS and OS among Pediatric ALL Patients

Risk factors selected for univariate analysis that had a statistically significant impact on pediatric ALL were included in the multivariate analysis. Finally, we identified that the SCCLG-ALL-2016 protocol (*HR* = 0.3, 95%*CI*: 0.1–0.5, *p* < 0.001), WBC ≥ 50 × 10^9^/L (*HR* = 2.3, 95%*CI*: 1.4–3.7, *p* < 0.001), PLT ≥ 20 × 10^9^/L (*HR* = 0.6, 95%*CI*: 0.4–1.0, *p* = 0.045), FFP transfusion (*HR* = 1.1, 95%*CI*: 1.0–1.1, *p* = 0.02), and D15 BM (*HR* = 1.7, 95%*CI*: 1.1–2.6, *p* = 0.014) were the independent factors for RFS (Table 3). As shown in Table 3, FFP transfusion was significantly associated with poor RFS (*HR* = 2.3, 95%*CI*: 1.2–4.4, *p* = 0.01), and D15-BM revealed a significant decrease in OS (*HR* = 2.1, 95%*CI*: 1.3–3.6, *p* = 0.005). Interestingly, for ABT, FFP transfusion was the unique independent factor that affected both RFS and OS.

Further comparison of the K-M survival curves of 196 patients with FFP transfusion and 553 patients without FFP transfusion was performed. The result indicated that the RFS and OS were significantly lower in those with FFP than in those without FFP (10 y RFS: 76.64 ± 8.3% vs. 84.98 ± 8.8%, *p* = 0.045;10 y OS: 81.8 ± 8.5% vs. 91.65 ± 9.3%, *p* < 0.001) (Figure 1A,B).

### 3.4. The Clinical Implication of Average FFP Transfusion among Pediatric ALL Patients

In the present study, the FFP transfusions varied from 0 to 2600 mL, with a mean of 126.8. Moreover, the average FFP transfusion was between 0 and 116.1 mL/kg, with a mean of 6.3 mL/kg. To determine the optimal threshold of average FFP transfusion, we initially assessed the average FFP transfusion as a continuous variable for risk of relapse from study entry.

To explore the relationship between average FFP transfusion and risk of relapse in ALL patients, we performed a fitting curve analysis by using the restrictive cubic spline function following Cox proportional hazards models. This analysis was performed using log-converted and unconverted data. Logarithms (relative risk) can be converted to relative risk by pairing logarithms. After adjusting for factors that may be associated with risk of relapse, including gender, chemotherapy protocol, risk group, D15-BM, D33-BM, D15-MRD, D33-MRD, CNSL, immunophenotype, WBC group, age group, ETV6/RUNX1 status, KMT2A status, and BCR/ABL1 status, average FFP transfusion was found to have a biphasic distribution (Figure 2), with an initial rapid increase in the low average FFP transfusion (the value of average FFP transfusion <25 mL/kg), while this was followed by a gradual linear decrease in the high average FFP transfusion greater than 25 mL/kg. Furthermore, the risk of relapse-free seemed to be reduced with increasing average FFP transfusion after the turning point (average FFP transfusion ≥25 mL/kg).

To more accurately identify the cut-off value of 25 mL/kg for average FFP transfusion that can distinguish patients with FFP transfusion at high risk from the low-risk patients, we performed threshold effect analysis using Cox proportional hazards models. The threshold effect of average FFP transfusion on risk of relapse was significant after adjusting for potential confounders. The adjusted regression coefficient (Log RR) was 1.2 (95%*CI*: 1.1–1.3, *p* = 0.002) for average FFP transfusion ≥25 mL/kg and 1.0 (95%*CI*: 1.0–1.1, *p* = 0.081) for average FFP transfusion <25 mL/kg (Table 4). Thus, average FFP transfusion of 25 mL/kg was selected as the prognostic threshold for further comparison. High average FFP transfusion was defined as ≥25 mL/kg, and low average FFP transfusion was defined as <25 mL/kg.

Subsequently, K-M survival analysis was performed to determine whether the average FFP transfusion of 25 mL/kg was suitable to confirm a prognostic threshold for classifying FFP transfusion populations into those at high risk for treatment failure and those who were not. The corresponding RFS at 10 years for those with high and low average FFP transfusion was 74.09 ± 7.8% and 82.02 ± 8.6% (*p* = 0.027; Figure 1C), respectively. Overall survival at 10 years for patients with high and low average FFP transfusion was 81.48 ± 8.4% and 90.09 ± 9.1%, respectively (*p* = 0.033; Figure 1D). This study confirmed our findings that the average FFP transfusion of 25 mL/kg is a clinically useful risk identification threshold in the FFP transfusion population.

## 4. Discussion

The immunosuppressive effects caused by ABT in recipients are termed TRIM. TRIM can produce adverse clinical effects. For example, the immune function of receptors is inhibited and the immune monitoring function is decreased [17], so TRIM can promote cancer progression in theory. At present, pediatric ALL patients, especially those in the intensive induction phase, are still the major indications for ABT. In our study, 96.6% of the childhood ALL patients received ABT, which was close to the 90–98.3% reported in the previous literature [11,12,18].

The early work on TRIM in childhood ALL was performed by Freiberg et al. [18]. They retrospectively analyzed 358 pediatric ALL patients from 1984 to 1988 during the induction phase and excluded 16 patients who received the most blood products. They concluded that the number of transfusions was largely an epiphenomenon that reflected disease severity and chemotherapy reduction during induction therapy, and a TRIM effect appeared unlikely to contribute strongly to outcome in childhood ALL. Subsequently, Alkayed et al. [14] also analyzed the ABT results of 136 childhood ALL patients and demonstrated that PRBC, SDP, and FFP transfusions did not have any significant association with relapse or death, so their conclusion was consistent with Freiberg et al.

However, Jaime-Pérez et al. [11] retrospectively studied 108 pediatric ALL patients between 2000 and 2009 for possible TRIM during the induction phase. Their results showed that the number of blood products transfused to children with ALL appeared to be significantly associated with lower survival rates. Even after the exclusion of outliers, the data still supported a TRIM effect. The authors concluded that TRIM effects may be an independent prognostic factor among pediatric ALL patients. Recently, Wang et al. [12] published their results regarding TRIM effects in 163 childhood patients with ALL between 2006 and 2011. In the cox regression analysis, more than 25 transfusions was predictive of death (*p* = 0) and relapse (*p* = 0.006). Thus, their conclusions were similar to those of Jaime-Pérez et al. [11], and they considered that TRIM might play an important part in the outcome of childhood ALL.

In this study, our univariate and multivariate analysis demonstrated that the SCCLG-ALL-2016 protocol, WBC ≥ 50 × 10^9^/L, PLT ≥ 20 × 10^9^/L, and D15 BM were independent factors for RFS. Moreover, D15-BM was also an independent factor for OS. As is well known, the above influencing factors have been clearly related to the prognosis of children with ALL in many previous literature reports [15,16,19,20,21,22,23,24]. Apart from these known independent factors, in terms of ABT, the transfusion of PRBCs and SDPs during the induction phase failed to show any independent effect on the clinical outcome of pediatric ALL. Interestingly, we found the transfusion of FFP was the unique independent factor that affected both RFS and OS. Furthermore, our survival analysis revealed that patients with FFP transfusion were identified to have inferior RFS and OS rates compared with those without FFP. Based on these results, we believe ABT may confer a TRIM effect in childhood ALL and seems likely to lead to the adverse clinical outcome. Unfortunately, both our univariate and multivariate analysis showed that the number of transfusions was not an independent prognostic factor, which was different from the conclusions of Jaime-Pérez et al. [11] and Wang et al. [12].

With regard to the difference between ours and the previous four studies, we implied that one of the explanations might be the preparation method used for the blood products. As far as we know, the blood products in the studies by Freiberg et al. [18] and by Alkayed et al. [14] were leukocyte-reduced and irradiated, whereas only leukocyte-reduced products were used in the studies by Jaime-Pérez et al. [11], Wang et al. [12], and ours. Irradiation might prevent mononuclear cell proliferation, which is necessary for TRIM. Secondly, the sample sizes vary greatly, which may contribute to the different conclusions. There is no doubt that our conclusion was more reliable, as it was based on a large sample.

In our retrospective analysis, the results also showed that patients who received FFP transfusion had a higher age, and it occurred more often in elder children. To illustrate this problem, we have to mention the indications of FFP transfusion among ALL children during the induction phase. During the induction period, the most common chemotherapeutic pharmaceutical was L-ASP, which causes fibrinogen decline and abnormal coagulation function. Therefore, most of the transfusions of FFP were actually to treat the adverse reactions to L-ASP. Previous studies have reported that elder age at diagnosis was associated with a higher frequency of the toxicities of L-ASP. This might be partly explained by the more aggressive therapy children older than 10 years of age typically receive. These associations remained after adjusting for treatment risk group, but biological differences across ages also appear to impact susceptibility [25,26]. Also, patients with FFP transfusion mostly received the SCCLG-ALL-2016 protocol, with more than 25 transfusions and more PRBC transfusions. Most importantly, our study suggested that the higher positive rate of D33-MRD, the elevation of the relapse rate, and the reduction in the survival rate were caused by the transfusion of FFP. We implied one of the reasons was related to TRIM, because ABTs were related to transfusion-associated immunosuppressive effects. Another critical reason was that FFP contains free asparagine and may replace the plasma asparagine pool during L-ASP therapy, and consequently, repletion of the asparagine pool with FFP administration during L-ASP therapy, leading to a certain antagonizing effect of L-ASP action, was possible [27].

Based on our findings, we hypothesize that average FFP transfusion may play an important role in the prognosis in pediatric ALL. Subsequently, we examined the risk of relapse of patients with different average FFP transfusions, which was performed using generalized additive models with the aim of confirming the optimal threshold of average FFP transfusion that was observed in such a large sample of pediatric ALL patients. Finally, the turning points were selected through piece-wise linear regression. Interestingly, as a result, average FFP transfusion of 25 mL/kg was confirmed as the prognostic threshold. This was further confirmed through K-M survival analysis, and average FFP transfusion more than 25 mL/kg indeed demonstrated inferior RFS and OS. Previous studies have shown that administration of FFP has no significant influence on the incidence of venous thromboembolism, hemorrhages, and the plasma level of antithrombin and fibrinogen during the induction phase in pediatric ALL patients [28,29,30,31,32]. Thus, we did not recommend frequent use of FFP during the L-ASP treatment period, and the total amount of FFP should be controlled and kept below 25 mL/kg. In addition, cryoprecipitation or fibrinogen might be a better option for the side-effect of L-ASP.

We have to acknowledge several limitations in our study. Firstly, due to the retrospective design, we cannot prove the relationship between cause and effect. Secondly, the transfusion decision was not standardized but mainly depended on the medical judgment of the pediatric physician; this may represent a possible source of bias. Thirdly, our data analysis did not consider the storage time of blood products transfused; this factor has been suggested to have a tumor-promoting effect. In spite of these limitations, to the best of our knowledge, this is the largest-scale study about ABT and the prognosis of pediatric ALL, representing fourteen years of single-center experience in the real world. Most importantly, we proposed the relationship between the transfusion threshold of FFP and the clinical outcome of childhood ALL for the first time, which has great guiding significance for clinical blood transfusion [30].

In summary, among the blood products, only FFP supplementation is closely related to the prognosis of childhood ALL. During the intensive induction phase, the indications of FFP transfusion should be strictly grasped, and the total amount of FFP should be controlled and kept below 25 mL/kg.

## Figures and Tables

**Figure 1 cancers-15-04462-f001:**
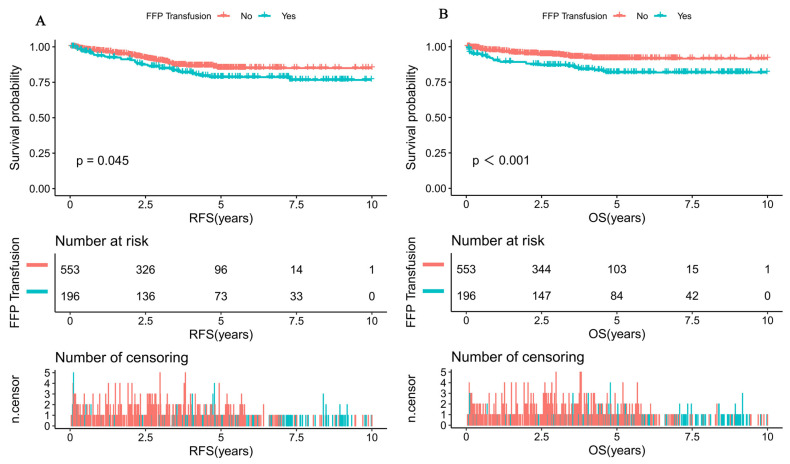
Survival curves of pediatric ALL patients according to whether patients received FFP transfusion and the threshold of average FFP transfusion. (**A**) Probability of RFS for patients with FFP transfusion and without FFP transfusion. (**B**) Probability of OS for patients with FFP transfusion and without FFP transfusion. (**C**) Probability of RFS for patients with different average FFP transfusions. (**D**) Probability of OS for patients with different average FFP transfusions.

**Figure 2 cancers-15-04462-f002:**
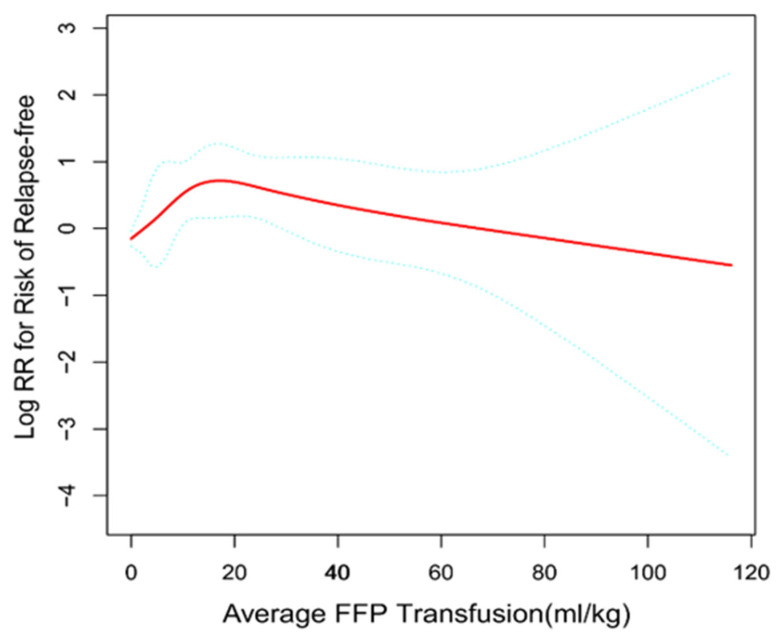
General additive models demonstrate the relationship between average FFP transfusion and the risk of relapse-free. The resulting figures show the predicted log (relative risk) on the *y*-axis and the average FFP transfusion on the *x*-axis.

**Table 1 cancers-15-04462-t001:** Baseline characteristics of study participants by FFP transfusion classification.

Characteristics	Total	FFP Transfusion		*p* Value
		No (*n* = 553)	Yes (*n* = 196)	
Gender, n (%)				0.483
Male	462 (61.7%)	337 (60.9%)	125 (63.8%)	
Female	287 (38.3%)	216 (39.1%)	71 (36.2%)	
Age (y), median (range)	4.4 (0.4–14.9)	4.1 (0.4–14.9)	5.7 (1.0–14.7)	<0.001
Age group (y)				<0.001
<10	634 (84.6%)	484 (87.5%)	150 (76.5%)	
≥10	115 (15.4%)	69 (12.5%)	46 (23.5%)	
Chemotherapy protocol, n (%)				<0.001
SCCLG-ALL-2016 Protocol	340 (45.4%)	247 (44.7%)	162 (82.7%)	
GD-ALL-2008 Protocol	409 (54.6%)	306 (55.3%)	34 (17.3%)	
Initial WBC (×10^9^/L), median (range)	9.3 (0.1–1095.0)	9.6 (0.1–1095.0)	8.6 (0.2–516.2)	0.608
WBC group, n(%)				0.574
<50 × 10^9^/L	610 (81.4%)	453 (81.9%)	157 (80.1%)	
≥50 × 10^9^/L	139 (18.6%)	100 (18.1%)	39 (19.9%)	
Initial Hb (g/L), median (range)	79.0 (24.0–172.0)	78.0 (24.0–149.0)	81.5 (28.0–172.0)	0.136
Hb group				0.097
<70 g/L	242 (32.3%)	188 (34.0%)	54 (27.6%)	
≥70 g/L	507 (67.7%)	365 (66.0%)	142 (72.4%)	
Initial PLT(×10^9^/L), median (range)	59.0 (2.0–634.0)	59.0 (2.0–634.0)	62.0 (8.0–417.0)	0.953
PLT group				0.091
<20 × 10^9^/L	116 (15.5%)	93 (16.8%)	23 (11.7%)	
≥20 × 10^9^/L	633 (84.5%)	460 (83.2%)	173 (88.3%)	
Risk group, n (%)				0.717
SR	154 (20.6%)	110 (19.9%)	44 (22.4%)	
IR	331 (44.2%)	248 (44.8%)	83 (42.3%)	
HR	264 (35.2%)	195 (35.3%)	69 (35.2%)	
Blood type				0.220
O	302 (40.3%)	227 (41.0%)	75 (38.3%)	
A	225 (30.0%)	173 (31.3%)	52 (26.5%)	
B	179 (23.9%)	122 (22.1%)	57 (29.1%)	
AB	43 (5.7%)	31 (5.6%)	12 (6.1%)	
Number of transfusions				<0.001
No transfusions	25 (3.3%)	24 (4.3%)	1 (0.5%)	
1 to 10 transfusions	555 (74.1%)	439 (79.4%)	116 (59.2%)	
11 to 25 transfusions	160 (21.4%)	87 (15.7%)	73 (37.2%)	
More than 25 transfusions	9 (1.2%)	3 (0.5%)	6 (3.1%)	
Transfusion of blood products				
PRBCs (U), median (range)	4.0 (0.0–25.5)	3.0 (0.0–25.5)	4.0 (0.0–21.0)	<0.001
SDPs (U), median (range)	1.0 (0.0–20.0)	1.0 (0.0–20.0)	1.0 (0.0–20.0)	0.105
Immunophenotype, n (%)				0.126
B	681 (90.9%)	509 (92.0%)	172 (87.8%)	
T	59 (7.9%)	37 (6.7%)	22 (11.2%)	
Biphenotypic	9 (1.2%)	7 (1.3%)	2 (1.0%)	
CNSL, n (%)				0.329
Yes	32 (4.3%)	26 (4.7%)	6 (3.1%)	
No	717 (95.7%)	527 (95.3%)	190 (96.9%)	
BCR/ABL1 status, n (%)				0.937
Negative	558 (94.6%)	415 (94.5%)	143 (94.7%)	
Positive	32 (5.4%)	24 (5.5%)	8 (5.3%)	
KMT2A status, n (%)				0.845
Negative	549 (96.8%)	416 (96.7%)	133 (97.1%)	
Positive	18 (3.2%)	14 (3.3%)	4 (2.9%)	
ETV6/RUNX1 status, n (%)				0.893
Negative	471 (83.2%)	360 (83.3%)	111 (82.8%)	
Positive	95 (16.8%)	72 (16.7%)	23 (17.2%)	
Prednisone response, n (%)				0.393
PGR	667 (90.1%)	490 (89.6%)	177 (91.7%)	
PPR	73 (9.9%)	57 (10.4%)	16 (8.3%)	
D15 BM, n (%)				0.062
M1	492 (66.0%)	374 (67.9%)	118 (60.5%)	
M2/M3	254 (34.0%)	177 (32.1%)	77 (39.5%)	
D33 BM, n (%)				0.085
M1	707 (96.6%)	532 (97.3%)	175 (94.6%)	
M2/M3	25 (3.4%)	15 (2.7%)	10 (5.4%)	
D15 MRD, n (%)				0.353
<0.1%	170 (30.0%)	138 (29.2%)	32 (34.0%)	
≥0.1%	396 (70.0%)	334 (70.8%)	62 (66.0%)	
D33 MRD, n (%)				0.013
<0.01%	476 (81.8%)	410 (83.5%)	66 (72.5%)	
≥0.01%	106 (18.2%)	81 (16.5%)	25 (27.5%)	

**Table 2 cancers-15-04462-t002:** Univariate analysis for RFS and OS among pediatric patients with ALL.

Variables	RFS		OS	
	*HR* (95%*CI*)	*p* Value	*HR* (95%*CI*)	*p* Value
Gender				
Male	Ref.		Ref.	
Female	0.7 (0.4, 1.1)	0.113	0.9 (0.5, 1.4)	0.542
Age group				
<10	Ref.		Ref.	
≥10	1.6 (0.9, 2.6)	0.098	3.3 (2.0, 5.6)	<0.001
Chemotherapy protocol				
GD-ALL-2008 Protocol	Ref.		Ref.	
SCCLG-ALL-2016 Protocol	0.3 (0.2, 0.5)	<0.001	0.0 (0.0, Inf)	0.995
WBC group				
<50 × 10^9^/L	Ref.		Ref.	
≥50 × 10^9^/L	2.4 (1.5, 3.8)	<0.001	2.3 (1.4, 4.0)	0.002
Hb group				
<70 g/L	Ref.		Ref.	
≥70 g/L	1.2 (0.7, 1.9)	0.468	1.2 (0.7, 2.1)	0.556
PLT group				
<20 × 10^9^/L	Ref.		Ref.	
≥20 × 10^9^/L	0.5 (0.3, 0.8)	0.008	0.6 (0.3, 1.1)	0.108
Risk group				
SR	Ref.		Ref.	
IR	1.0 (0.6, 1.7)	0.937	1.6 (0.7, 3.4)	0.229
HR	1.1 (0.6, 2.0)	0.937	2.5 (1.2, 5.2)	0.019
Blood type				
O	Ref.		Ref.	
A	1.3 (0.8, 2.2)	0.260	1.0 (0.6, 1.8)	0.985
B	1.0 (0.6, 1.7)	0.939	0.9 (0.5, 1.7)	0.747
AB	1.0 (0.4, 2.5)	0.943	0.5 (0.1, 2.0)	0.302
Number of transfusions				
No transfusions	Ref.		Ref.	
1 to 10 transfusions	1.0 (0.3, 3.1)	0.945	1.8 (0.3, 13.4)	0.548
11 to 25 transfusions	1.3 (0.4, 4.4)	0.643	3.9 (0.5, 28.6)	0.187
More than 25 transfusions	1.1 (0.1, 10.7)	0.926	6.3 (0.6, 69.6)	0.133
PRBCs (U), median (range)	1.1 (1.0, 1.1)	0.004	1.1 (1.0, 1.2)	<0.001
SDPs (U), median (range)	1.1 (1.0, 1.1)	0.087	1.1 (1.0, 1.2)	0.002
FFP transfusion				
No	Ref.		Ref.	
Yes	1.5 (1.1, 2.4)	0.047	2.4 (1.4, 3.9)	<0.001
Immunophenotype				
B	Ref.		Ref.	
T	1.5 (0.7, 2.9)	0.283	2.3 (1.2, 4.6)	0.015
Biphenotypic	0.9 (0.1, 6.8)	0.952	1.5 (0.2, 11.0)	0.679
CNSL				
Yes	Ref.		Ref.	
No	1.4 (0.3, 5.6)	0.654	2.3 (0.3, 16.4)	0.415
BCR/ABL1 status				
Positive	Ref.		Ref.	
Negative	0.8 (0.3, 2.7)	0.764	1.8 (0.3, 13.3)	0.550
KMT2A status, n (%)				
Positive	Ref.		Ref.	
Negative	9316259.7 (0.0, Inf)	0.995	0.5 (0.1, 2.1)	0.345
ETV6/RUNX1 status				
Positive	Ref.		Ref.	
Negative	1.9 (0.8, 4.8)	0.172	1.2 (0.5, 3.1)	0.718
Prednisone response				
PGR	Ref.		Ref.	
PPR	1.1 (0.6, 2.2)	0.747	1.8 (0.9, 3.6)	0.081
D15 BM				
M1	Ref.		Ref.	
M2/M3	2.0 (1.3, 3.1)	<0.001	2.1 (1.3, 3.5)	0.002
D33 BM				
M1	Ref.		Ref.	
M2/M3	1.0 (0.2, 4.1)	0.997	4.0 (1.6, 10.0)	0.003
D15 MRD				
<0.1%	Ref.		Ref.	
≥0.1%	1.4 (0.7, 2.8)	0.349	2.4 (0.8, 7.1)	0.103
D33 MRD				
<0.01%	Ref.		Ref.	
≥0.01%	1.6 (0.8, 3.1)	0.194	2.2 (0.9, 5.4)	0.079

**Table 3 cancers-15-04462-t003:** Multivariate analysis for RFS and OS among pediatric patients with ALL.

Outcome	Variable	*HR* (95%*CI*)	*p* Value
RFS	SCCLG-ALL-2016 Protocol	0.3 (0.1, 0.5)	<0.001
	WBC ≥ 50 × 10^9^/L	2.3 (1.4, 3.7)	<0.001
	PLT ≥ 20 × 10^9^/L	0.6 (0.4, 1.0)	0.045
	PRBCs	1.0 (0.6, 1.6)	0.937
	FFP transfusion	1.1 (1.0, 1.1)	0.020
	D15 BM	1.7 (1.1, 2.6)	0.014
OS	Age ≥10 y	1.3 (0.7, 2.3)	0.342
	WBC ≥ 50 × 10^9^/L	1.8 (1.0, 3.3)	0.071
	HR	0.9 (0.4, 2.2)	0.790
	PRBCs	1.1 (1.0, 1.2)	0.153
	SDPs	1.0 (0.9, 1.1)	0.788
	FFP	2.3 (1.2, 4.4)	0.010
	D15 BM	2.1 (1.3, 3.6)	0.005
	D33 BM	2.4 (0.8, 6.6)	0.102

**Table 4 cancers-15-04462-t004:** Threshold effect analysis of the association between average FFP transfusion and risk of relapse using piece-wise linear regression.

Average FFP Transfusion	Log*RR* (95%CI)	*p* Value
<25 mL/kg	1.0 (1.0, 1.1)	0.081
≥25 mL/kg	1.2 (1.1, 1.3)	0.002

Adjusted for gender, chemotherapy protocol, risk group, BM blast, peripheral blood blasts, CNSL, FAB category, karyotype, WBC group, age group, CEBPA status, WT1 status, and NPM1 status.

## Data Availability

The data sets used and/or analyzed in the current study are available from the corresponding author on reasonable request.

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
