# Peer review of "Real-World Presentation and Prognostic Effect of Allogeneic Blood Transfusion during the Intensive Induction Phase in Pediatric Acute Lymphoblastic Leukemia"

_cancers, 2023, doi:10.3390/cancers15184462_

Round 1

Reviewer 1 Report

Aim of this study was to determine associations between ABT during the intensive induction phase of therapy and prognostic effect in a real-world cohorts of pediatric patients with ALL. A total of 749 children were enrolled, treated from February 2008 to May 2022. Two successive protocols were utilized but the induction phase was similar and consisted of VCR+Dexamethasone+Anthracycline+ L-Asparaginase. As usual, during induction phase, patients need packed reed cells and platelets. Fresh frozen plasma (FFP) is selectively transfused in patients with altered coagulation profile. In this cohort of pediatric patients, 196 of them received FFP.

In the multivariate analysis, FFP transfusion was the unique independent factor that affected both RFS and OS. Other risk factors selected for the univariate analysis that had a statistically significant impact on pediatric ALL were included in the multivariate analysis. The authors identified that SCCLG-ALL-2016 protocol, WBC ≥50×109/L, PLT ≥20×109/, D15 BM, n addition to FFP transfusion were independent factor for RFS.

I have the following comments:

1.     The alteration of the coagulation profile which led to the request for FFP, may be due to the drug L-Asparaginase, notoriously hepatotoxic. Even if, we know that FFP contains free asparagine, I would like to know if patients with abnormal coagulation function, requiring FFP, received fewer doses of L-asparaginase compared to those who, not having toxicity, continued the therapy regularly. in fact, the authors report that adolescents (age > 10 years), in which asparaginase toxicity is more common, received FFP more frequently than younger children.

2.     Children were treated with two protocols. The authors reported that drug classes and the composition of the chemotherapeutic protocols, at the intensive induction phase, were essentially the same. Since the protocol SCCLG-ALL-2016 was an independent risk factor for RFS, I would like to know the differences between the two protocols.

However, in this paper data support that TRIM is an independent prognostic factor among pediatric ALL.

The manuscript is well written and the statistical analysis is detailed and correctly conducted.

Author Response

Comments 1:  The alteration of the coagulation profile which led to the request for FFP, may be due to the drug L-Asparaginase, notoriously hepatotoxic. Even if, we know that FFP contains free asparagine, I would like to know if patients with abnormal coagulation function, requiring FFP, received fewer doses of L-asparaginase compared to those who, not having toxicity, continued the therapy regularly. in fact, the authors report that adolescents (age > 10 years), in which asparaginase toxicity is more common, received FFP more frequently than younger children.

Response 1: Thank you for pointing this out. Actually, patients with abnormal coagulation function, requiring FFP, received equal doses of L-asparaginase compared to those who, not having toxicity, continued the therapy regularly in our study. Elder children may be more sensitive to the toxicity of L-ASP, so an increase in FFP is required.

Comments 2:   Children were treated with two protocols. The authors reported that drug classes and the composition of the chemotherapeutic protocols, at the intensive induction phase, were essentially the same. Since the protocol SCCLG-ALL-2016 was an independent risk factor for RFS, I would like to know the differences between the two protocols.

Response 2: We apologize for not clarifying this and it confused you. The differences between the two protocols listed as follows. Firstly, include MRD in the evaluation criteria in terms of risk level. Secondly, in the IR group, VCR/PEG-ASP was added for one week after CAM, and PEG-ASP was used 7 times throughout the entire course of treatment; Increase CTX on the tenth day of T-ALL induced chemotherapy.

Reviewer 2 Report

While this paper is well-written, there are necessary revisions and additions in response to the comments provided below:

1. Data Source: The study mentions that it's a single-centre retrospective cohort study. This inherently introduces the potential for center-specific biases. How generalizable are these findings to other centers or regions? 

2. Multivariate Analysis: The paper states that the transfusion of FFP was significantly associated with various factors. Were potential confounders considered in the analysis? Factors such as disease severity or other treatment modalities could influence the requirement for FFP and thereby prognosis.

3. Cut-off Value Determination: The methodology used for determining the cut-off value for FFP transfusion is mentioned. However, is there any clinical rationale behind this specific cut-off, or is it purely based on statistical findings?

4. Comparative Analysis: The abstract mentions that only FFP supplementation is closely related to the prognosis of childhood ALL. However, was there a direct comparison made between the groups receiving different types of transfusions? For instance, was there a group that did not receive any transfusion to compare against?

5. Sample Size: Only 196 (26.2%) received FFP transfusion. Given this smaller subgroup, is the sample size sufficient for drawing robust conclusions especially when analyzing further subcategories?

 6. Causal Relationship: The study establishes an association between FFP transfusion and prognosis, but how certain are the authors that this is a causal relationship? Could it be possible that the children who needed more FFP had a different disease trajectory to begin with, rather than the FFP influencing the prognosis?

7. Clinical Implications: The conclusions suggest strict control on FFP transfusion. Are there specific guidelines or alternative treatments proposed for children who might need FFP transfusion above the identified threshold?

Author Response

Comments 1:  Data Source: The study mentions that it's a single-centre retrospective cohort study. This inherently introduces the potential for center-specific biases. How generalizable are these findings to other centers or regions? 

Response 1: Thanks for your useful suggestion. Although this study is a single center retrospective cohort study, our center belongs to the leader unit of the South China Children's Leukemia Collaborative Group. We led the implementation of two protocols, GD-ALL-2008 and SCCLG-ALL-2016. Currently, more than 20 hospitals in South China have used this protocol, and due to the consistent chemotherapy regimen used, this study is scalable and applicable to more than 20 other centers.

Comments 2:   Multivariate Analysis: The paper states that the transfusion of FFP was significantly associated with various factors. Were potential confounders considered in the analysis? Factors such as disease severity or other treatment modalities could influence the requirement for FFP and thereby prognosis.

Response 2:We agree with this comment. In our multivariate analysis, we have considered relevant confounding factors, including age, gender, disease severity, treatment protocol etc. After adjusting for confounding factors, we conducted a multivariate analysis. Therefore, our statistical analysis is reasonable and correct.

Comments 3:  Cut-off Value Determination: The methodology used for determining the cut-off value for FFP transfusion is mentioned. However, is there any clinical rationale behind this specific cut-off, or is it purely based on statistical findings?

Response 3: Thank you for pointing this out. The cut-off value determination is purely based on statistical findings in our study data. Please see Page 9, line 219-232.

Comments 4:   Comparative Analysis: The abstract mentions that only FFP supplementation is closely related to the prognosis of childhood ALL. However, was there a direct comparison made between the groups receiving different types of transfusions? For instance, was there a group that did not receive any transfusion to compare against?

Response 4: We are sorry for not expressing this issue clearly. In fact, we mentioned in Tables 2 that RFS and OS statistics were conducted using no blood transfusion as a reference value. Please see Page 6, Table 2.

Comments 5:  Sample Size: Only 196 (26.2%) received FFP transfusion. Given this smaller subgroup, is the sample size sufficient for drawing robust conclusions especially when analyzing further subcategories?

Response 5:Thank you very much for your suggestion, which enabled us to make good revisions to our manuscript. In fact, statistics believe that over 100 cases belong to a large sample, so we believe that 196 cases are sufficient to draw a stable conclusion. Secondly, statistical analysis of more than 30 subgroups showed clinical significance.

 Comments 6:  Causal Relationship: The study establishes an association between FFP transfusion and prognosis, but how certain are the authors that this is a causal relationship? Could it be possible that the children who needed more FFP had a different disease trajectory to begin with, rather than the FFP influencing the prognosis?

Response 6: We used multivariate analysis after adjusting for confounding factors to establish a clear relationship between FFP infusion and prognosis. For some patients who initially require a lot of FFP infusion, confounding factors were also included in the adjustment, so this conclusion is reliable.

Comments 7:   Clinical Implications: The conclusions suggest strict control on FFP transfusion. Are there specific guidelines or alternative treatments proposed for children who might need FFP transfusion above the identified threshold?

Response 7: For those who might need FFP transfusion above the identified threshold,  cryoprecipitation or fibrinogen might be a better option for the side-effect of L-ASP. Please see Page 12, Line 358-359.